# Tuning nonlinear damping in graphene nanoresonators by parametric–direct internal resonance

Ata Keşkekler [1✉], Oriel Shoshani[2], Martin Lee[3], Herre S. J. van der Zant [3], Peter G. Steeneken [1,3] & Farbod Alijani [1✉]

Mechanical sources of nonlinear damping play a central role in modern physics, from solid-state physics to thermodynamics. The microscopic theory of mechanical dissipation suggests that nonlinear damping of a resonant mode can be strongly enhanced when it is coupled to a vibration mode that is close to twice its resonance frequency. To date, no experimental evidence of this enhancement has been realized. In this letter, we experimentally show that nanoresonators driven into parametric-direct internal resonance provide supporting evidence for the microscopic theory of nonlinear dissipation. By regulating the drive level, we tune the parametric resonance of a graphene nanodrum over a range of 40–70 MHz to reach successive two-to-one internal resonances, leading to a nearly two-fold increase of the nonlinear damping. Our study opens up a route towards utilizing modal interactions and parametric resonance to realize resonators with engineered nonlinear dissipation over wide frequency range.

[1] Department of Precision and Microsystems Engineering, Delft University of Technology, Mekelweg 2, Delft 2628 CD, The Netherlands. [2] Department of Mechanical Engineering, Ben-Gurion University of Negev, Beersheba, Israel. [3] Kavli Institute of Nanoscience, Delft University of Technology, Lorentzweg 1, Delft 2628 CJ, The Netherlands. ✉email: a.keskekler-1@tudelft.nl; f.alijani@tudelft.nl

In nature, from macro- to nanoscale, dynamical systems evolve towards thermal equilibrium while exchanging energy with their surroundings. Dissipative mechanisms that mediate this equilibration convert energy from the dynamical system of interest to heat in an environmental bath. This process can be intricate, nonlinear, and in most cases hidden behind the veil of linear viscous damping, which is merely an approximation valid for small amplitude oscillations.

In the last decade, nonlinear dissipation has attracted much attention with applications that span nanomechanics[1], materials science[2], biomechanics[3], thermodynamics[4], spintronics[5], and quantum information[6]. It has been shown that the nonlinear dissipation process follows the empirical force model $F_d = -\tau_{nl1}x^2\dot{x}$ where $\tau_{nl1}$ is the nonlinear damping coefficient, $x$ is the displacement, and $\dot{x}$ velocity. To date, the physical mechanism from which this empirical damping force originates has remained ambiguous, with a diverse range of phenomena being held responsible including viscoelasticity[7], phonon–phonon interactions[8,9], Akheizer relaxation[10], and mode coupling[11]. The fact that nonlinear damping can stem from multiple origins simultaneously makes isolating one route from the others a daunting task, especially since the nonlinear damping coefficient $\tau_{nl1}$ is perceived to be a fixed parameter that unlike stiffness[12–14], quality factor[15], and nonlinear stiffness[16–18], cannot be tuned easily.

Amongst the different mechanisms that affect nonlinear damping, intermodal coupling is particularly interesting, as it can be enhanced near internal resonance (IR), a special condition at which the ratio of the resonance frequencies of the coupled modes is a rational number[19]. This phenomenon has frequently been observed in nano/micromechanical resonators[20–29]. At IR, modes can interact strongly even if their nonlinear coupling is relatively weak. Interestingly, IR is closely related to the effective stiffness of resonance modes, and can therefore be manipulated by careful engineering of the geometry of mechanical systems, their spring hardening nonlinearity[30,31], and electrostatic spring softening[29]. IR also finds its route in the microscopic theory of dissipation proposed back in 1975, where it was hypothesized to lead to a significantly shorter relaxation time if there exists a resonance mode in the vicinity of twice the resonance frequency of the driven mode in the density of states[32].

Here, we demonstrate that nonlinear damping of graphene nanodrums can be strongly enhanced by parametric–direct IR, providing supporting evidence for the microscopic theory of nonlinear dissipation[10,32]. To achieve this, we bring the fundamental mode of the nanodrum into parametric resonance at twice its resonance frequency, allowing it to be tuned over a wide frequency range from 40 to 70 MHz. We extract the nonlinear damping as a function of the parametric drive level, and observe that it increases as much as 80% when the frequency shift of the parametric resonance brings it into IR with a higher mode. By comparing the characteristic dependence of the nonlinear damping coefficient on parametric drive to a theoretical model, we confirm that IR can be held accountable for the significant increase in nonlinear damping.

## Results

**Measurements.** Experiments are performed on a 10 nm thick multilayer graphene nanodrum with a diameter of 5 μm, which is transferred over a cavity etched in a layer of $SiO_2$ with a depth of 285 nm. A blue laser is used to thermomechanically actuate the membrane, where a red laser is being used to detect the motion, using interferometry (see "Methods" for details). A schematic of the setup is shown in Fig. 1a.

By sweeping the drive frequency, we obtain the frequency response of the nanodrum in which multiple directly driven resonance modes can be identified (Fig. 1b). We find the fundamental axisymmetric mode of vibration at $f_{0,1} = 20.1$ MHz and several other modes, of which the two modes, at $f_{2,1}^{(1)} = 47.4$ MHz and $f_{2,1}^{(2)} = 50.0$ MHz, are of particular interest. This is because, to study the effect of IR on nonlinear damping, we aim to achieve a 2:1 IR by parametrically driving the fundamental mode, such that it coincides with one of the higher frequency modes. The frequency ratios $f_{2,1}^{(1)}/f_{0,1} \approx 2.3$ and $f_{2,1}^{(2)}/f_{0,1} \approx 2.4$ are close to the factor 2, however additional frequency tuning is needed to reach the 2:1 IR condition.

The parametric resonance can be clearly observed by modulating the tension of the nanodrum at frequency $\omega_F$ with the blue laser while using a frequency converter in the vector network analyzer (VNA) to measure the amplitude at $\omega_F/2$ as shown in Fig. 1c. By increasing the parametric drive, we observe a Duffing-type geometric nonlinearity over a large frequency range, such that the parametrically driven fundamental resonance can be tuned across successive 2:1 IR conditions with modes $f_{2,1}^{(1)}$ and $f_{2,1}^{(2)}$, respectively.

In Fig. 1c, we observe that the parametric resonance curves follow a common response until they reach the saddle-node bifurcation frequency $f_{SNB}$ above which the parametric resonance curve reaches its peak amplitude $A_{SNB}$ and drops down to low amplitude. We note that the value of $A_{SNB}$ can be used to determine the degree of nonlinear damping[33]. Therefore, to extract the nonlinear damping coefficient $\tau_{nl1}$ of mode $f_{0,1}$ from the curves in Fig. 1c, we use the following single-mode model to describe the system dynamics

$$\ddot{x}_1 + \omega_1^2 x_1 + \gamma x^3 = F_1 x_1 \cos(\omega_F t) - 2\tau_1 \dot{x}_1 - 2\tau_{nl1}x_1^2\dot{x}_1, \quad (1)$$

in which $\omega_1 = 2\pi f_{0,1}$ is the eigenfrequency of the axisymmetric mode of the nanodrum, $\gamma$ is its Duffing constant, and $F_1$ and $\omega_F$ are the parametric drive amplitude and frequency, respectively. Moreover, $2\tau_1 = \omega_1/Q$ is the linear damping coefficient, with $Q$ being the quality factor, and $\tau_{nl1}$ is the nonlinear damping term of van der Pol type that prevents the parametric resonance amplitude $A_{SNB}$ from increasing to infinity[33,34] at higher driving frequencies since $|A_{SNB}|^2 \propto (2F_1Q - 4)/\tau_{nl1}$. To identify the parameters governing the device dynamics from the measurements in Fig. 1c, we use Eq. (1) and obtain good fits of the parametric resonance curves using $\tau_{nl1}$ and $\gamma$ as fit parameters (see Supplementary Note I).

As we gradually increase the drive level, $f_{SNB}$ increases until it reaches the vicinity of the IR, where we observe an increase in $\tau_{nl1}$ (Fig. 1d). Whereas $f_{SNB}$ increases with parametric drive $F_1$, Fig. 1c shows that its rate of increase $\frac{df_{SNB}}{dF_1}$ slows down close to $f_{2,1}^{(1)}$, locking the saddle-node bifurcation frequency when $f_{SNB} \approx 45$ MHz. At the same time, $\tau_{nl1}$ increases significantly at the associated parametric drive levels, providing the possibility to tune nonlinear damping up to twofolds by controlling $F_1$, as seen in Fig. 1d.

Figure 1c also shows that above a certain critical parametric drive level $F_{1,crit}$, the frequency locking barrier at $f_{SNB} \approx 45$ MHz is broken and $f_{SNB}$ suddenly jumps to a higher frequency ($\approx 5$ MHz higher), and a corresponding larger $A_{SNB}$. We label this increase in the rate $\frac{df_{SNB}}{dF_1}$ by "surge" in Fig. 1c, where an abrupt increase in the amplitude–frequency response is observed to occur above a critical drive level $F_{1,crit}$. Interestingly, even above $F_{1,crit}$ a further increase in $\tau_{nl1}$ is observed with increasing drive amplitude, indicating that a similar frequency locking occurs when the parametric resonance peak reaches the second IR at $f_{SNB} \approx f_{2,1}^{(2)}$. Similar nonlinear phenomena are also showcased in a second nanodrum undergoing parametric–direct modal interaction,

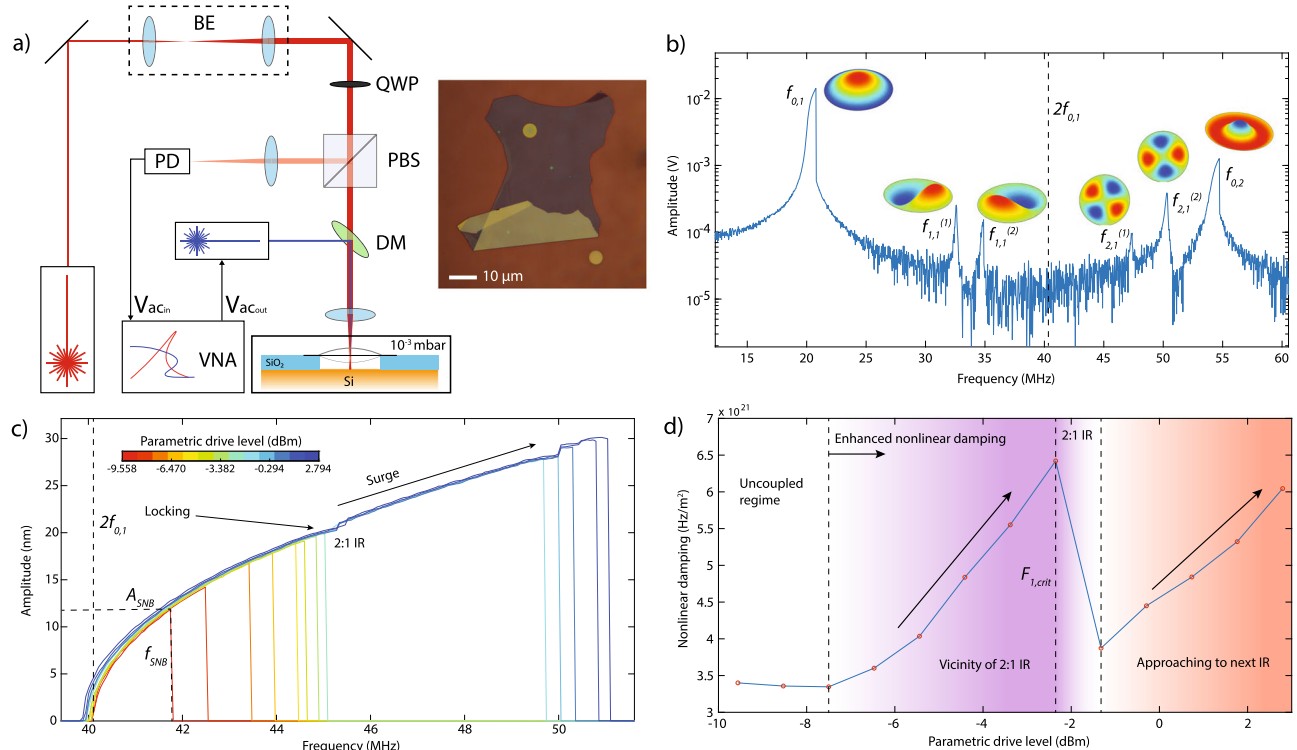

**Fig. 1 Nonlinear dynamic response of a graphene nanodrum near 2:1 internal resonance. a** Fabry–Pérot interferometry with thermomechanical actuation and microscope image of the graphene. Experiments are performed in vacuum at $10^{-3}$ mbar. Red laser is used to detect the motion of the graphene drum and the blue laser is used to optothermally actuate it. BE beam expander, QWP quarter wave plate, PBS polarized beam splitter, PD photodiode, DM dichroic mirror, VNA vector network analyzer, $V_{ac_{in}}$ analyzer port, $V_{ac_{out}}$ excitation port. In the device schematic, Si and SiO$_2$ layers are represented by orange and blue colors, respectively. **b** Direct frequency response curve of the device (motion amplitude vs. drive frequency), showing multiple resonances (drive level $= -12.6$ dBm). The mode shapes are simulated by COMSOL. Resonance peaks are associated with $f_{m,n}^{(k)}$ where $m$ represents the number of nodal diameters, $n$ nodal circles, and $k = 1, 2$ stand for the first and second asymmetric degenerate modes. Dashed line shows $2f_{0,1}$, which is the drive frequency where the parametric resonance of mode $f_{0,1}$ is activated. **c** Parametric resonance curves (calibrated motion amplitude vs. drive frequency), driven at twice the detection frequency. As the parametric resonance curves approach the 2:1 internal resonance (IR), $f_{SNB}$ first locks to 2:1 IR frequency and consecutively saddle-node bifurcation surges to a higher frequency and amplitude. $A_{SNB}$ and $f_{SNB}$ stand for the amplitude and frequency of saddle-node bifurcation. **d** Variation of the nonlinear damping $\tau_{nl1}$ as a function of drive $F_1$. Dashed lines represent different regimes of nonlinear damping. White region represents a constant nonlinear damping, purple region an increase in nonlinear damping in the vicinity of 2:1 IR and orange region an increase in nonlinear damping due to IR with a higher mode.

confirming the reproducibility of the observed physics (see Supplementary Note II).

**Theoretical model.** Although the single-mode model in Eq. (1) can capture the response of the parametric resonance, it can only do so by introducing a nonphysical drive level dependent nonlinear damping coefficient $\tau_{nl1}(F_1)$ (Fig. 1d). Therefore, to study the physical origin of our observation, we extend the model by introducing a second mode whose motion is described by generalized coordinate $x_2$. Moreover, to describe the coupling between the interacting modes at the 2:1 IR, we use the single term coupling potential $U_{cp} = \alpha x_1^2 x_2$ (see Supplementary Note III). The coupled equations of motion in the presence of this potential become

$$\ddot{x}_1 + \omega_1^2 x_1 + \gamma x_1^3 + \frac{\partial U_{cp}}{\partial x_1} = F_1 x_1 \cos(\omega_F t) - 2\tau_1 \dot{x}_1 - 2\tau_{nl1} x_1^2 \dot{x}_1,$$

$$\ddot{x}_2 + \omega_2^2 x_2 + \frac{\partial U_{cp}}{\partial x_2} = F_2 \cos(\omega_F t) - 2\tau_2 \dot{x}_2. \tag{2}$$

The two-mode model describes a parametrically driven mode with generalized coordinate $x_1$ coupled to $x_2$ that has

eigenfrequency $\omega_2 = 2\pi f_{2,1}^{(1)}$, damping ratio $\tau_2$, and is directly driven by a harmonic force with magnitude $F_2$.

To understand the dynamics of the system observed experimentally and described by the model in Eq. (2), it is convenient to switch to the rotating frame of reference by transforming $x_1$ and $x_2$ to complex amplitude form (see Supplementary Note IV). This transformation reveals a system of equations that predicts the response of the resonator as the drive parameters ($F_1$, $F_2$, and $\omega_F$) are varied. Solving the coupled system at steady state yields the following algebraic equation for the amplitude $a_1$ of the first mode

$$\left[\tau_1 + (\tau_{nl1} + \tilde{\alpha}^2 \tau_2)\frac{a_1^2}{4}\right]^2 + \left[\Delta\omega_1 - \left(\frac{3\gamma}{\omega_F} + \tilde{\alpha}^2 \Delta\omega_2\right)\frac{a_1^2}{4}\right]^2$$
$$= \frac{1}{4\omega_F^2}\left[F_1^2 + \tilde{\alpha}^2(F_2^2 + 2\omega_F \Delta\omega_2 F_1 F_2/\alpha)\right], \tag{3}$$

where $\Delta\omega_1 = \omega_F/2 - \omega_1$ and $\Delta\omega_2 = \omega_F - \omega_2$ are the frequency detuning from the primary and the secondary eigenfrequencies, and $\tilde{\alpha}^2 = \alpha^2/[\omega_F^2(\tau_2^2 + \Delta\omega_2^2)]$ is the rescaled coupling strength. Essentially, the first squared term in Eq. (3) captures the effect of damping on the parametric resonance amplitude $a_1$, the second term captures the effect of nonlinear coupling on the stiffness and driving frequency, and the term on the right side is the effective

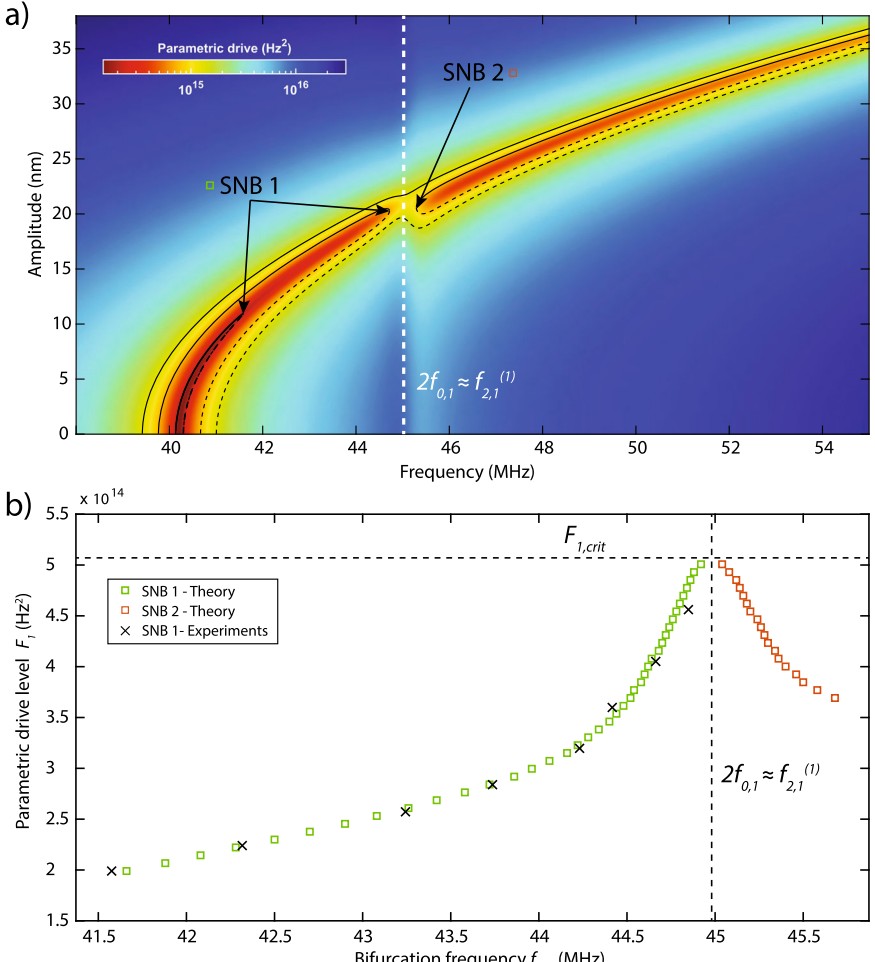

**Fig. 2 Parametric–direct internal resonance. a** Color map of the analytical model response curves obtained by using the fitted parameters from experiments. Colors correspond to frequency response (motion amplitude vs. drive frequency) solutions with a certain parametric drive level. Black lines show samples from these solutions where solid lines are stable and dashed lines are unstable solutions. White dashed line is where parametric resonance meets with interacting mode and undergoes internal resonance. **b** The underlying route of the amplitude–frequency surge is revealed by tracing the evolution of saddle-node bifurcations (green and red squares represent theoretical SNB1 and SNB2, whereas experimental SNB1 is represented by crosses) of the parametric resonance curves.

parametric drive. From the rescaled coupling strength $\tilde{\alpha}$ and Eq. (3) it can be seen that the coupling $\tilde{\alpha}^2$ shows a large peak close to the 2:1 IR where $|\Delta\omega_2| \approx 0$. In addition, Eq. (3) shows that mode 2 will always dissipate energy from mode 1 once coupled, and that the two-mode model accounts for an increase in the effective nonlinear damping parameter ($\tau_{nleff} = \tau_{nl1} + \tilde{\alpha}^2\tau_2$) near IR, in accordance with the observed peak in $\tau_{nl1}$ with the single-mode model in Fig. 1d. It is also interesting to note that this observation in steady state is different from what has been reported in Shoshani et al.[24] for transient nonlinear free vibrations of coupled modes where it was important that $\tau_2 \gg \tau_1$ to observe nonlinear damping. The two-mode model of Eq. (3) allows us to obtain good fits of the parametric resonance curves in Fig. 1b, with a constant $\tau_{nleff} \approx 3.4 \times 10^{21}$ (Hz/m$^2$) determined far from IR and a single coupling strength $\alpha = 2.2 \times 10^{22}$ (Hz$^2$/m) which intrinsically accounts for the variation of $\tau_{nleff}$ near IR. These fits can be found in Supplementary Note V, and demonstrate that the two-mode model is in agreement with the experiments for constant parameter values, without requiring drive level dependent fit parameters. We note that the extracted nonlinear damping parameter fits the Duffing response at $f_{0,1}$ with good accuracy too (see Supplementary Note VI).

To understand the physics associated with the frequency locking and amplitude–frequency surge, we use the experimentally extracted fit parameters from the two-mode model and numerically generate parametric resonance curves using Eq. (3) for a large range of drive amplitudes (see Fig. 2a). We see that for small drive levels, an upward frequency sweep will follow the parametric resonance curve and then will lock and jump down at the first saddle-node bifurcation (SNB1) frequency, which lies close to $f_{SNB} \approx f_{2,1}^{(1)}$. At higher parametric drive levels, the parametric resonance has a stable path to traverse the IR toward a group of stable states at higher frequencies.

A more extensive investigation of this phenomenon can be carried out by performing bifurcation analysis of the steady-state solutions (see Supplementary Note IV). The bifurcation analysis reveals two saddle-node bifurcations near the singular region of the IR, one at the end of the first path (SNB1) and another at the beginning of the second path (SNB2) (Fig. 2b). As the drive amplitude increases, the bifurcation pair starts to move toward each other until they annihilate one another to form a stable solution at the connecting point, which we labeled as "surge." It is also possible to observe that the rate at which saddle-node pairs approach each other dramatically drops near the IR condition,

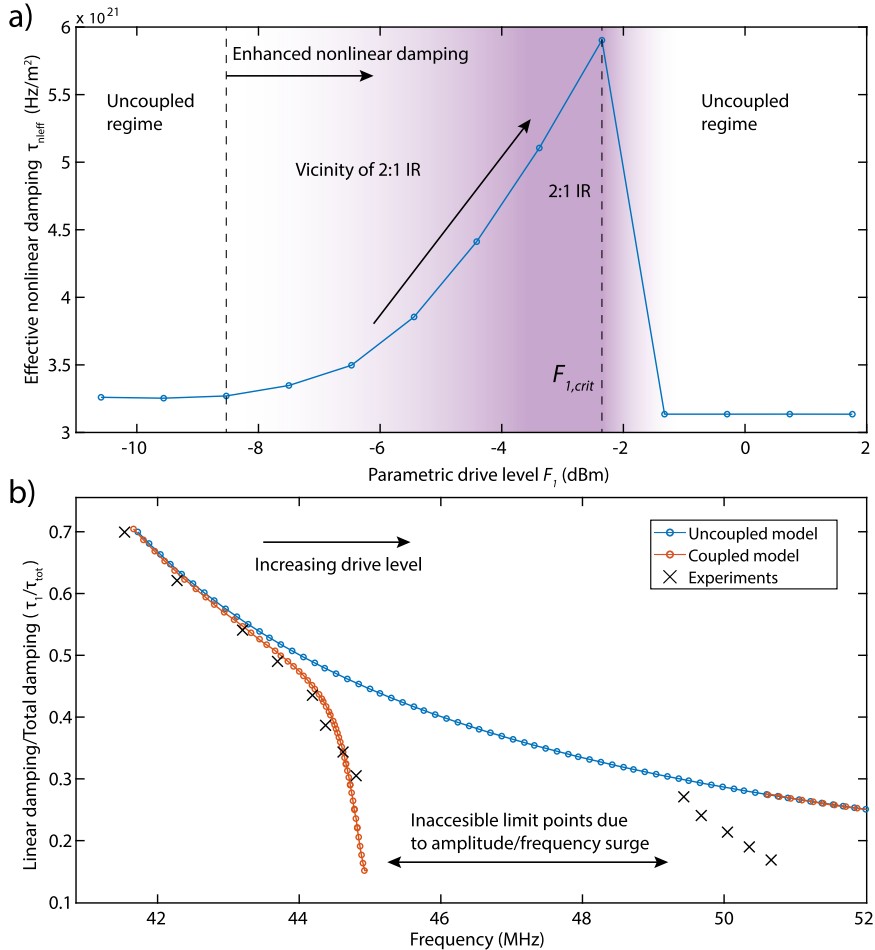

**Fig. 3 Measurements and fits of the effective nonlinear damping. a** Variation of the effective nonlinear damping parameter ($\tau_{nleff}$) with respect to parametric drive. The $\tau_{nleff}$ is obtained by fitting the numerically generated curves of Fig. 2a as the fit parameter. Dashed lines represent different regimes of nonlinear damping. White regions represent a constant nonlinear damping and purple region represents an increase in nonlinear damping in the vicinity of 2:1 IR. **b** Comparison of the ratio between linear damping ($\tau_1$) and total damping ($\tau_{tot}$). In the figure, blue and red dashed lines represent $\tau_1/\tau_{tot}$ obtained from uncoupled and coupled models, whereas black crosses represent experiments.

demonstrating the "locking" which we also observed in the experiments.

To check how closely the two-mode model captures the variation of $\tau_{nl1}$ close to the IR condition, we follow a reverse path, and fit the numerically generated resonance curves of Fig. 2a using the single-mode model of Eq. (3) with $\tau_{nl1}$ as the fit parameter. In this way, we track the variation of $\tau_{nl1}$ in the single-mode model with the parametric drive $F_1$, similar to what we observed experimentally and reported in Fig. 1c. The result of this fit is shown in Fig. 3a, where a similar anomalous change of nonlinear damping is obtained for the two-mode model.

The variation of nonlinear damping affects the total damping (sum of linear and nonlinear dissipation) of the resonator too. It is of interest to study how large this effect is. In Fig. 3b, we report the variation in the ratio of the linear damping $\tau_1$ and the amplitude-dependent total damping $\tau_{tot} = (\omega_1/Q + 0.25\tau_{nleff}|x_1|^2)$[33] in the spectral neighborhood of $f_{2,1}^{(1)}$, and observe a sudden decrease in the vicinity of IR. This abrupt change in the total damping is well captured by the two-mode model. With the increase in the drive amplitude, $\tau_1/\tau_{tot}$ values of this model though deviate from those of the experiments due to a subsequent IR at $f_{2,1}^{(2)}/f_{0,1} \approx 2.4$ that is not included in our theoretical analysis. The dependence of $\tau_1/\tau_{tot}$ on frequency shows that near IR, the total damping of the fundamental mode increases nearly by 80%. We note that

increased nonlinear damping near IR was also observed in Güttinger et al.[11]. In that work, nonlinear damping was studied using ringdown measurements, with two modes brought close to an IR by electrostatic gating. The increased nonlinear damping was attributed to a direct–direct 3:1 IR, which as shown theoretically in Shoshani et al.[24] leads to a high order (quintic) nonlinear damping term. Conversely, in our work, two modes are brought into parametric–direct 2:1 IR by adjusting the parametric drive level. This results in a nonlinear damping term that already comes into play at smaller amplitudes because it is of lower (cubic) order, as discussed in Shoshani et al.[24]. Moreover, the nonlinear damping mechanism in Güttinger et al.[11] is approximately described by two exponential decays with crossovers from $(\tau_1 + \tau_2)/2$ to $\tau_1$, which implies that similar to Shoshani et al.[24], $\tau_2 > \tau_1$ is required to observe positive nonlinear damping. This is in contrast with the damping mechanism we describe, where the effective nonlinear damping actually increases for smaller $\tau_2$ (see Eq. (3)).

## Discussion

Since the tension of the nanodrum can be manipulated by laser heating, we can further investigate the tunability of the nonlinear damping by increasing the laser power and detecting the range over which 2:1 IR conditions may occur. When increasing the blue laser power and modulation, we observe the parametrically

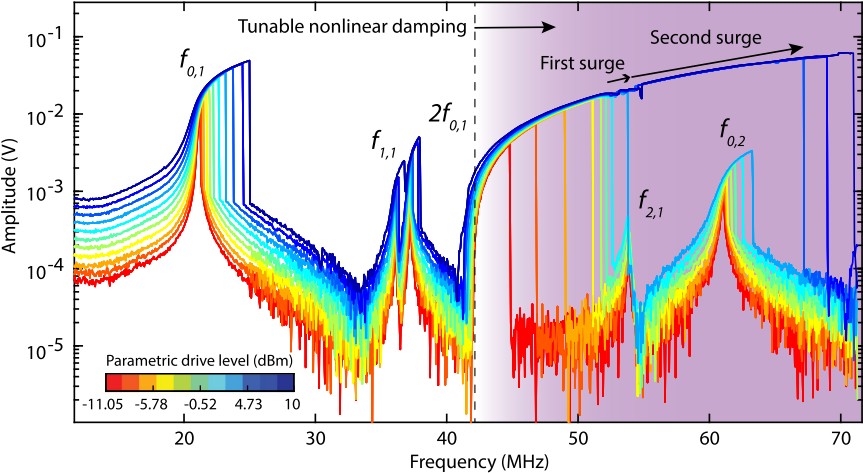

**Fig. 4 Nonlinear frequency response measurements at high drive powers.** The parametric resonance interacts successively with multiple directly driven modes of vibration. The arrows in the figure show successive amplitude–frequency surges. Starting from the dashed line, shaded area represents the region where nonlinear damping is tunable.

actuated signal also in the direct detection mode (like in Fig. 1b) due to optical readout nonlinearities[35]. As a result a superposition of Fig. 1b, c is obtained, as shown in Fig. 4. We note that the enhanced laser power increases membrane tension which moves $f_{0,1}$ upward by a few MHz, but also allows us to reach even higher parametric modulation. In this configuration, we achieve a frequency shift in $f_{SNB}$ from 40 to 70 MHz, corresponding to as much as 75% tuning of the mechanical motion frequency. This large tuning can increase the number of successive IRs that can be reached even further, to reach modal interactions between the parametric mode $f_{0,1}$ and direct modes $f_{2,1}^{(2)}$ and $f_{0,2}$ (see Fig. 4). As a result, multiple amplitude–frequency surges can be detected in the large frequency range of 30 MHz over which nonlinear damping coefficient can be tuned.

In summary, we study the tunability of nonlinear damping in a graphene nanomechanical resonator, where the fundamental mode is parametrically driven to interact with a higher mode. When the system is brought near a 2:1 IR, a significant increase in nonlinear damping is observed. In addition, the rate of increase of the parametric resonance frequency reduces in a certain locking regime, potentially stabilizing the values of $f_{SNB}$ and $A_{SNB}$, which could potentially aid frequency noise reduction[21]. Interestingly, as the drive level is further increased beyond the critical level $F_{1,crit}$, this locking barrier is broken, resulting in a surge in $f_{SNB}$ and amplitude of the resonator. These phenomena were studied experimentally, and could be accounted for using a two-mode theoretical model. The described mechanism can isolate and differentiate mode coupling induced nonlinear damping from other dissipation sources, and sheds light on the origins of nonlinear dissipation in nanomechanical resonators. It also provides a way to controllably tune nonlinear damping which complements existing methods for tuning linear damping[15], linear stiffness[12–14] and nonlinear stiffness[16–18], extending our toolset to adapt and study the rich nonlinear dynamics of nanoresonators.

## Methods

**Sample fabrication.** Devices are fabricated using standard electron-beam (e-beam) lithography and dry etching techniques. A positive e-beam resist (AR-P-6200) is spin coated on a Si wafer with 285 nm of thermally grown $SiO_2$. The cavity patterns ranging from 2 to 10 μm in diameter are exposed using the Vistec EBPG 5000+ and developed. The exposed $SiO_2$ are subsequently etched away in a reactive ion etcher using a mixture of $CHF_3$ and Ar gas until all the $SiO_2$ is etched away and the Si exposed. Graphene flakes are then exfoliated from natural crystal and dry transferred on top of cavities.

**Laser interferometry.** The experiments are performed at room temperature in a vacuum chamber ($10^{-3}$ mbar). A power modulated blue laser ($\lambda = 405$ nm) is used to thermomechanically actuate the nanodrum. The motion is then readout by using a red laser ($\lambda = 633$ nm) whose reflected intensity is modulated by the motion of the nanodrum in a Fabry–Pérot etalon formed by the graphene and the Si back mirror (Fig. 1a). The reflected red laser intensity from the center of the drum is detected using a photodiode, whose response is read by the same VNA that modulates the blue laser. The measured VNA signal is then converted to displacement in nanometers using a nonlinear optical calibration method[35] (see Supplementary Note VII).

## Data availability

The data that support the findings of this study are available from the corresponding authors upon request.

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

## Acknowledgements

The authors would like to thank Prof. Marco Amabili for fruitful discussions about nonlinear damping. The research leading to these results received funding from European Union's Horizon 2020 research and innovation program under Grant Agreement 802093 (ERC starting grant ENIGMA). O.S. acknowledges support for this work from the United States–Israel Binational Science Foundation under Grant No. 2018041. P.G.S. and H.S.J.v.d.Z. acknowledge funding from the European Union's Horizon 2020 research and innovation program under grant agreement numbers 785219 and 881603 (Graphene Flagship).

## Author contributions

A.K., O.S., H.S.J.v.d.Z., P.G.S., and F.A. conceived the experiments; A.K. fabricated the graphene samples and conducted the measurements; M.L. fabricated the chips with cavities; O.S. built the theoretical model; O.S. and A.K. performed the fitting; A.K., O.S., P.G.S. and F.A. did data analysis and interpretation; F.A. supervised the project; and the manuscript was written by A.K. and F.A. with inputs from all authors.

## Competing interests

The authors declare no competing interests.
