## [Peer Review File · Nature Communications]

Reviewers' Comments:

Reviewer #1:

Remarks to the Author:

In their manuscript "Tuning nonlinear damping in nanoresonators by parametric-direct internal resonance", the authors A. Keskekler et al. study experimentally and theoretically a mechanical resonator with coupled modes. When one of the modes has exactly twice the resonance frequency of another mode, an increase of the effective nonlinear damping is observed through the frequency of a bifurcation point. The authors achieve an explanation of this phenomenon in terms of the nonlinear coupling between the involved modes. The model matches their experimental data quite precisely.

The paper is well structured and written in a clear way. The conclusions and the methodology are correct in all aspects that I can judge, and the work can be reproduced on the basis of the provided data. This is an interesting manuscript that meets the technical quality requirements of Nature Communications (with one caveat, see point A below). However, I believe it will have a reduced impact because it follows the findings of a different study (see point B below).

A: My only scientific criticism concerns the amount of available data. Leaving aside the "overview" sweeps in Fig. 1b and Fig. 4, the entire paper rests on a set of 13 frequency sweeps with increasing drive strength. Given the complexity of the model, I wonder why the authors did not acquire more data. For instance, it should have been easy to initialize the resonator beyond the 2:1 frequency position and thereby access the second saddle-node bifurcation for low driving strength. There is also no verification of the phenomenon with a second sample, which would make the study yet more convincing. For a high-impact journal, the small amount of data is a weakness.

B: The paper is similar in content to a previous work that the authors cite as Ref. [11]. There, coupled modes with a 3:1 frequency ratio led to an increase of nonlinear damping. The model is not identical, but the previous paper does of course reduce the novelty of the finding.

Reviewer #2:

Remarks to the Author:

This is an interesting study on nonlinear damping in mechanical resonators. The authors show that the nonlinear damping can be tuned by sweeping the frequency of the vibrating mode. Nonlinear damping becomes particularly large when the vibrating mode is in internal resonance with another mode.

Nonlinear damping is an important effect in nanomechanics. However, the microscopic origin remains unclear in most of the devices. Here, the experiment shows how nonlinear damping emerges from internal resonance. This is a compelling result. This work may be published in a high-impact journal such as Nature Communications, if the authors answer the following questions in a convincing way.

1-The authors show increased nonlinear damping with the peak in Fig. 1d. But is it clear that the peak is related to the nonlinear damping force? The authors fit the data with Eq 1 which includes nonlinear damping but not internal resonance.

2-In the text after Eq. 3, the authors give a compact relation between the nonlinear damping coefficient and the mode coupling. How is this relation derived for the 1:2 internal resonance? Why does internal resonance lead to nonlinear damping? Does this happen when the linear damping of mode 2 is much larger than that of mode 1, as discussed for the 1:3 internal resonance in the paper of Dykman (Ref. 24) ?

3-Did you measure the nonlinear damping with another method when the system is not in internal resonance? This would be useful to check whether the method used to extract the nonlinear damping in Fig. 1d is solid. For instance, the nonlinear damping can be extracted from resonantly driven vibrations in the nonlinear Duffing regime, as discussed in Ref. 33.

4-In Fig. 1b, the resonance frequencies of the higher modes are 47.4 MHz and 50 MHz. Why do the authors associate the peak in the nonlinear damping in Fig. 1d to a higher mode, since the peak occurs around 45 MHz in Fig. 1c?

5-The authors write: "To identify the parameters governing the device dynamics from the measurements in Fig. 1c, we use Eq. (1) and obtain good fits of the parametric resonance curves (see Supplementary Note IV)." However, it is difficult to judge whether the fits are good or not. There are different curves in the figure of the Supplementary Notes, and they are not labelled.

Here are less important questions

6-Can you discuss the similarities & differences of your measurements in Fig. 1c with that in Figs. 1b,c of the paper C. Chen, D. H. Zanette, D. A. Czaplewski, S. Shaw, and D. Lopez, "Direct observation of coherent energy transfer in nonlinear micromechanical oscillators," Nature communications, vol. 8, p. 15523, 2017.

7-The authors write: "It is also possible to observe that the rate at which saddle-node pairs approach each other dramatically drops near the internal resonance condition, demonstrating the "locking" which we also observed in the experiments." Please show the data, at least in the Supplementary Notes.

8-The authors write " In addition, the rate of increase of the parametric resonance frequency reduces in a certain locking regime, stabilizing the values of f_{SNB} and AS_{NB} , which could potentially aid frequency noise reduction" Is this observed in you experiment? If not, explain why?

Reviewer #3:

Remarks to the Author:

Report on the manuscript entitled

"Tuning nonlinear damping in nanoresonators by parametric-direct internal resonance"

submitted for publication to Nature Communications

by

A. Keskekler, O. Shoshani, M. Lee, H. S. J. van der Zant, P. G. Steeneken, and F. Alijani

In this manuscript the authors present experimental results of nonlinear damping tuning for nanoresonators using parametric-direct internal resonance (as announced by the title)and reduced order modeling of the phenomenon.

The manuscript is clearly written. Main results are presented in 7 pages and Supplementary informations are provided (5 Supplementary Notes).

The noteworthy results are :

- To my knowledge, design of interactions is still an original track in mechanical systems. It is much more original in the field of nano-electro-mechanical systems and significant to the field and potential applications (mass sensing, etc.).

- I am not aware of similar results in the literature.

Nonlinear damping has already been seen experimentally in acoustic membranes used as Nonlinear Energy Sinks but not described in nano-resonators with such experimental evidences.

The agreement between results obtained via reduced order models with one or two modes and experimental results is noticeable. The identification of parameters from experiments to derive theoretical / analytical calculations and to obtain the results of the model is well described and the experimental process is sound.

Analytical methods are clearly described in the Supplementary notes and support the analysis of experiments.

The work (data, their analysis, comparison theory/experiments and identification process) supports the conclusions and claims (but Supplementary Information are needed to carry out additional evidence to the 7 pages).

I recommend a minor revision before publication.

It is not necessary that I would review again the revised version of this manuscript.

Minor remarks

1) Clarify experimental/fitted results in FIG.5 (Supplementary Note V.) and stable/unstable/fitted curves.

2) Marginal question : For the two last values of parametric excitation, isolas can be seen (out of the frequency bandwidth of interest). Does the model return similar results ?

3) Check notation $\tau_{n\text{leff}}$ page 3.

4) Page 2 - Supplementary Note II. The argument « we assume harmonic motion of the form [...] as a first approximation. » is correct but it could also be derived from normal form analysis.

Reviewer #1 (Remarks to the Author):

In their manuscript “Tuning nonlinear damping in nanoresonators by parametric direct internal resonance”, the authors A. Keskekler et al. study experimentally and theoretically a mechanical resonator with coupled modes. When one of the modes has exactly twice the resonance frequency of another mode, an increase of the effective nonlinear damping is observed through the frequency of a bifurcation point. The authors achieve an explanation of this phenomenon in terms of the nonlinear coupling between the involved modes. The model matches their experimental data quite precisely.

The paper is well structured and written in a clear way. The conclusions and the methodology are correct in all aspects that I can judge, and the work can be reproduced on the basis of the provided data. This is an interesting manuscript that meets the technical quality requirements of Nature Communications (with one caveat, see point A below). However, I believe it will have a reduced impact because it follows the findings of a different study (see point B below).

We thank the referee for reviewing our manuscript and giving us opportunity to clarify.

1-) My only scientific criticism concerns the amount of available data. Leaving aside the “overview” sweeps in Fig. 1b and Fig. 4, the entire paper rests on a set of 13 frequency sweeps with increasing drive strength. Given the complexity of the model, I wonder why the authors did not acquire more data. For instance, it should have been easy to initialize the resonator beyond the 2:1 frequency position and thereby access the second saddle-node bifurcation for low driving strength. There is also no verification of the phenomenon with a second sample, which would make the study yet more convincing. For a high-impact journal, the small amount of data is a weakness.

We thank the referee for this comment. Indeed, as requested, we have acquired more data on a second sample (2.5 \$\mu\text{m}\$ radius – 14 nm thick) exhibiting a similar parametric-direct internal resonance. We provide these additional datasets in Supplementary Note II, strengthening support for the reproducibility of the observed phenomena.

The suggestion to try to access the second saddle-node bifurcation is very interesting, but not easy, because it is a so-called isola. Accessing such isolas experimentally is an intricate task, since the isolated response is disconnected from the main solution path. To reach it, one would need to first initialize the system beyond the interaction regime and push the system to high amplitude state, e.g. by noise induced amplitude jumps, or by combining swept sine and stepped sine excitations [see for instance: Detroux T, Noël JP, Virgin LN, Kerschen G (2018) Experimental study of isolas in nonlinear systems featuring modal interactions. PLOS ONE 13(3): e0194452]. So although interesting, this task is beyond the scope of this paper which focuses on the tunability of the nonlinear damping.

2-) The paper is similar in content to a previous work that the authors cite as Ref. [11]. There, coupled modes with a 3:1 frequency ratio led to an increase of nonlinear damping. The model is not identical, but the previous paper does of course reduce the novelty of the finding.

We agree that both our paper and Ref. [11] show intermodal transfer of energy in a nanoresonator and discuss nonlinear damping. But, the reported nonlinear damping mechanism and the approach to the problem are significantly different. To further emphasize these differences and the novelty of the current work, we have added the following paragraph on page 4 before the discussion section:

“We note that increased nonlinear damping near internal resonance was also observed in [11]. In that work nonlinear damping was studied using ringdown measurements, with 2 modes brought close to an internal resonance by electrostatic gating. The increased nonlinear damping was attributed to a direct-direct 3:1 internal resonance that as shown theoretically in Ref. [24] leads to a high order (quintic) nonlinear damping term. Conversely, in our work two modes are brought into parametric-direct 2:1 internal resonance by adjusting the parametric drive level. This results in a nonlinear damping term that already comes into play at smaller amplitudes because it is of lower (cubic) order, as discussed in Ref. [24]. Moreover, the nonlinear damping mechanism in [11] is approximately described by two exponential decays with crossovers from $(\tau_1 + \tau_2)/2$ to τ_1 , which implies that similar to [24], $\tau_2 > \tau_1$ is required to observe positive nonlinear damping. This is in contrast with the damping mechanism we describe where the effective nonlinear damping actually increases for smaller τ_2 (see Eq. 3).“

Reviewer #2 (Remarks to the Author):

This is an interesting study on nonlinear damping in mechanical resonators. The authors show that the nonlinear damping can be tuned by sweeping the frequency of the vibrating mode. Nonlinear damping becomes particularly large when the vibrating mode is in internal resonance with another mode. Nonlinear damping is an important effect in nanomechanics. However, the microscopic origin remains unclear in most of the devices. Here, the experiment shows how nonlinear damping emerges from internal resonance. This is a compelling result. This work may be published in a high-impact journal such as Nature Communications, if the authors answer the following questions in a convincing way.

We thank the referee for reviewing our manuscript and giving us opportunity to clarify.

1-) The authors show increased nonlinear damping with the peak in Fig. 1d. But is it clear that the peak is related to the nonlinear damping force? The authors fit the data with Eq 1 which includes nonlinear damping but not internal resonance.

In Fig. 1d we used a model with no internal resonance to fit the experimental data. The outcome of our fitting was a drive level dependent nonlinear damping term which shows a maximum as the parametric resonance frequency is close to a higher frequency mode. To prove the observed effect is in fact due to an internal resonance, we analyze the system using a 2 mode model (Eq. 2) that can reproduce the observed internal resonance. The model obtains a similar peak (see Fig. 3a), and thus provides evidence that the observation in Fig. 1d is related to internal resonance.

2-) In the text after Eq. 3, the authors give a compact relation between the nonlinear damping coefficient and the mode coupling. How is this relation derived for the 1:2 internal resonance? Why does internal resonance lead to nonlinear damping? Does this happen when the linear damping of mode 2 is much larger than that of mode 1, as discussed for the 1:3 internal resonance in the paper of Dykman (Ref. 24)?

The derivation of the relation for 2:1 mode coupling can be found in the Supplementary Note III. The effect of nonlinear damping on amplitude due to mode coupling is expressed by the first squared term in Eq. 3. This term shows that the nonlinear damping coefficient $\tau_{nl\ eff}$ depends on the term $\tilde{\alpha}^2 \tau_2 = \alpha^2 \tau_2 / [\omega_F^2 (\tau_2^2 + \Delta\omega_2^2)]$. At internal resonance, $\Delta\omega_2^2$ has a minimum, such that $\tilde{\alpha}^2$ and thus nonlinear damping is maximal. In our work, nonlinear damping is dependent on τ_2 like in [24], however $\tau_2 \gg \tau_1$ is not essential for the nonlinear damping mechanism near internal resonance we describe here. We have further clarified these points on page 3 of the manuscript.

3-) Did you measure the nonlinear damping with another method when the system is not in internal resonance? This would be useful to check whether the method used to extract the nonlinear damping in Fig. 1d is solid. For instance, the nonlinear damping can be extracted from resonantly driven vibrations in the nonlinear Duffing regime, as discussed in Ref. 33.

We followed reviewer's suggestion and compared extracted nonlinear damping values of τ_{nl1} obtained with parametric drive far from internal resonance to those obtained from directly driven resonances at different driving levels. Good fits of all curves (both parametric and directly driven) can be obtained with a single value of τ_{nl1} , as shown in new data that we present in Supplementary Note VI, as was also demonstrated in Ref. [33]. The new fits show that the method we used to extract nonlinear damping from parametric resonance can be used with confidence.

4-) In Fig. 1b, the resonance frequencies of the higher modes are 47.4 MHz and 50 MHz. Why do the authors associate the peak in the nonlinear damping in Fig. 1d to a higher mode, since the peak occurs around 45 MHz in Fig. 1c?

We associate the peak in nonlinear damping to internal resonance between the fundamental mode and $f_{2,1}$ because the peak frequency of the parametric curve slows down and locks near this frequency, as is best visible in Fig. 4. We believe that the highest frequency at which we observe frequency locking in Fig. 1c is slightly below the frequency $f_{2,1}$ shown in Fig. 1b due to quite large steps taken in parametric drive level (in dBm) in Fig. 1c, such that we might have stepped over the actual drive level at which both frequencies align.

5-) The authors write: "To identify the parameters governing the device dynamics from the measurements in Fig. 1c, we use Eq. (1) and obtain good fits of the parametric resonance curves (see Supplementary Note IV)." However, it is difficult to judge whether the fits are good or not. There are different curves in the figure of the Supplementary Notes, and they are not labelled.

The fits are now labelled and mapped to the results in the main paper (See Supplementary Notes I and V).

6-) Can you discuss the similarities & differences of your measurements in Fig. 1c with that in Figs. 1b,c of the paper C. Chen, D. H. Zanette, D. A. Czaplewski, S. Shaw, and D. Lopez, "Direct observation of coherent energy transfer in nonlinear micromechanical oscillators," Nature communications, vol. 8, p. 15523, 2017.

In the work of Chen et. al [23], Fig. 1b,c display the frequency response of a directly driven MEMS resonator showing a hardening response, until the resonance peak reaches internal resonance f_{IR} and "locks" to that frequency. This phenomenon is associated with a very sharp and discrete saturation of the resonance frequency. In our measurements, a similar frequency locking happens, however it emerges progressively. Although this locking effect in [23] seems somewhat similar to the current work, it arises from a quite different modal interaction model. The tunability of nonlinear damping we discuss here is not demonstrated in [23].

7-) The authors write: "It is also possible to observe that the rate at which saddle node pairs approach each other dramatically drops near the internal resonance condition, demonstrating the "locking" which we also observed in the experiments." Please show the data, at least in the Supplementary Notes.

In the mentioned sentence, we are pointing at Fig. 2b, where it is possible to trace theoretically (depicted by squares in the graph) and experimentally (depicted by crosses in the graph) the saddle-node bifurcation points as a function of drive level. From the figure it is possible to observe that df_{SNB}/dF_1 gets smaller as the f_{SNB} approaches the internal resonance condition. The experimental points here are the jump-down frequencies of the parametric resonance curves for different drive levels.

8-) The authors write” In addition, the rate of increase of the parametric resonance frequency reduces in a certain locking regime, stabilizing the values of f_{SNB} and A_{SNB} , which could potentially aid frequency noise reduction” Is this observed in you experiment? If not, explain why?

Our reasoning for an increased stability of the values f_{SNB} and A_{SNB} can be inferred from Fig. 2b. In this figure as the peak of the resonance curve gets closer to the internal resonance condition, the sensitivity of f_{SNB} to the drive level drops dramatically, with df_{SNB}/dF_1 (a measure for the stability of f_{SNB}) being almost zero at the internal resonance point. So our experiments do show a higher stability of f_{SNB} towards variations in drive level F_1 .

We have not experimentally studied the effect of locking on the frequency noise in the system, as indicated by the word ‘potentially’. We expect that if the frequency noise is dominated by noise in $F_1(t)$, it will likely be reduced near the locking point (since $df_{SNB}/dF_1=0$), but for other noise sources this is less likely.

Reviewer #3 (Remarks to the Author):

Report on the manuscript entitled

"Tuning nonlinear damping in nanoresonators by parametric-direct internal resonance"

submitted for publication to Nature Communications

by A. Keskekler, O. Shoshani, M. Lee, H. S. J. van der Zant, P. G. Steeneken, and F. Alijani

In this manuscript the authors present experimental results of nonlinear damping tuning for nanoresonators using parametric-direct internal resonance (as announced by the title) and reduced order modeling of the phenomenon. The manuscript is clearly written. Main results are presented in 7 pages and Supplementary informations are provided (5 Supplementary Notes).

The noteworthy results are:

To my knowledge, design of interactions is still an original track in mechanical systems. It is much more original in the field of nano-electro-mechanical systems and significant to the field and potential applications (mass sensing, etc.).

I am not aware of similar results in the literature.

Nonlinear damping has already been seen experimentally in acoustic membranes used as Nonlinear Energy Sinks but not described in nano-resonators with such experimental evidences. The agreement between results obtained via reduced order models with one or two modes and experimental results is noticeable. The identification of parameters from experiments to derive theoretical / analytical calculations and to obtain the results of the model is well described and the experimental process is sound.

Analytical methods are clearly described in the Supplementary notes are support the analysis of experiments. The work (data, their analysis, comparison theory/experiments and identification process) supports the conclusions and claims (but Supplementary Information are needed to carry out additional evidence to the 7 pages). I recommend a minor revision before publication. It is not necessary that I would review again the revised version of this manuscript.

We would like to thank the referee for the recognition of our work and the constructive assessment. Moreover, we are thankful for the valuable comments regarding the current version of our manuscript.

Minor remarks

1-) Clarify experimental/fitted results in FiG.5 (Supplementary Note V.) and stable/unstable/fitted curves.

To clarify the relation between the main text and the fits in the Supplementary Notes, we have labelled and mapped the fitting results to the results in the main paper. (See Supplementary Notes I and V).

2-) Marginal question: For the two last values of parametric excitation, isolas can be seen (out of the frequency bandwidth of interest). Does the model return similar results?

Indeed, for very high excitation levels, we observe additional interactions beyond the frequencies of interest, where similar phenomena occur (additional surge and increase of nonlinear damping, which can also be seen in the orange region in Fig. 1d). However, the model cannot explain those additional interactions since it only accounts for 2 coupled modes.

3-) Check notation $\tau_{n\text{leff}}$ page 3.

This notation is now corrected.

4-) Page 2 - Supplementary Note II. The argument « we assume harmonic motion of the form [...] as a first approximation. » is correct but it could also be derived from normal form analysis.

We thank the reviewer for their suggestion and have made this clear in the Supplementary material which is now Supplementary note III.

Reviewers' Comments:

Reviewer #1:

Remarks to the Author:

The authors have addressed all of my concerns to my satisfaction. I recommend publication in Nature Communications without further revisions.

Reviewer #2:

Remarks to the Author:

This work demonstrates how nonlinear damping emerges from 2:1 mode coupling. Nonlinear damping is a phenomenon that is observed by many groups, but its microscopic origin is usually poorly understood. For this reason, this work is important for the nanomechanics community. The measurements and the analysis are convincing. The response of the authors to my comments allowed me to better understand their results. I recommend publication in Nature Communications.